Corrected: Author Correction

# Modularity increases rate of floral evolution and adaptive success for functionally specialized pollination systems

Agnes S. Dellinger [1]*, Silvia Artuso[2], Susanne Pamperl [1], Fabián A. Michelangeli[3], Darin S. Penneys[4], Diana M. Fernández-Fernández[5], Marcela Alvear[6], Frank Almeda[6], W. Scott Armbruster[7], Yannick Staedler[1] & Jürg Schönenberger [1]

Angiosperm flowers have diversified in adaptation to pollinators, but are also shaped by developmental and genetic histories. The relative importance of these factors in structuring floral diversity remains unknown. We assess the effects of development, function and evolutionary history by testing competing hypotheses on floral modularity and shape evolution in Merianieae (Melastomataceae). Merianieae are characterized by different pollinator selection regimes and a developmental constraint: tubular anthers adapted to specialized buzz-pollination. Our analyses of tomography-based 3-dimensional flower models show that pollinators selected for functional modules across developmental units and that patterns of floral modularity changed during pollinator shifts. Further, we show that modularity was crucial for Merianieae to overcome the constraint of their tubular anthers through increased rates of evolution in other flower parts. We conclude that modularity may be key to the adaptive success of functionally specialized pollination systems by making flowers flexible (evolvable) for adaptation to changing selection regimes.

[1] Department of Botany and Biodiversity Research, University of Vienna, Rennweg 14, 1030 Vienna, Austria. [2] Department of Biosciences, University of Salzburg, Hellbrunnerstraße 34, 5020 Salzburg, Austria. [3] Institute of Systematic Botany, The New York Botanical Garden, 2900 Southern Blvd, Bronx, NY 10458-5126, USA. [4] Department of Biology and Marine Biology, University of North Carolina Wilmington, 601S. College Road, Wilmington, NC 28403, USA. [5] Herbario Nacional del Ecuador (QCNE), Instituto Nacional de Biodiversidad, Río Coca E06-115 e Isla Fernandina, Quito, Ecuador. [6] Institute of Biodiversity Science and Sustainability, California Academy of Sciences, 55 Music Concourse Drive, San Francisco, CA 94118-4503, USA. [7] School of Biological Science, University of Portsmouth, King Henry 1 Street, Portsmouth P012DY, UK. *email: agnes.dellinger@univie.ac.at

Modularity, the relative independence of some trait clusters from others within an organism, is a pervasive concept of evolutionary biology[1,2]. A module itself is defined as a cluster of traits which are highly correlated, but show weak correlation to other such trait clusters[2]. Modularity may apply to and originate through various processes, including developmental pathways, genetic constraints/pleiotropy or functional relationships between traits[2–4]. In theory, modules may evolve independently of each other and can respond independently to changing selection regimes[5–10]. Hence, it has been proposed that modularity may increase an organism's ability to adapt to novel selection pressures, increasing its evolvability[5,6,11].

The relative importance of developmental and genetic histories and functional adaptations in shaping morphological diversity has been studied extensively in anthropology and zoology[1,9,10]. In animals, development is used most often to explain patterns of modularity[12]. Studies on modularity in plants are scarce, however, and we still lack a clear perspective on the role of modularity in the evolution of the diversity of flowers that arose over the past 140 my[7,12–19]. A recent review suggested that modularity in flowers may originate through other processes than in animals (i.e., through function rather than development[12],) and hence research on floral modularity has the potential to expand our existing concepts of shape evolution.

Flowers represent ideal systems to test hypotheses on modularity. They comprise different organ types, which arise through different developmental pathways[18,20], and may hence show strong developmental modularity. In addition, these organs may carry very different functions in the plant's reproductive process such as pollinator attraction and efficient pollen transfer, and thus possibly show functional modularity[11,21]. Furthermore, flowers of two closely related species, which are pollinated by different pollinator groups, may underlie very different selection regimes[11,21]. Surprisingly, only few studies have assessed whether and how differences in selection regimes affect intra-floral correlation structures and none of these studies has tested competing hypotheses on modularity on the entire 3-dimensional floral structure[15,16,21].

In this study, we present a novel approach to the study of flower shape evolution through the integration of advanced imaging techniques (High-Resolution X-ray Computed Tomography[22,23]), state-of-the-art landmark-based geometric morphometrics, and phylogenetic comparative methods[24]. We chose Merianieae (Melastomataceae) as study system for three reasons. First, the group is characterized by repeated independent shifts from an ancestral bee pollination syndrome to systems involving different vertebrate pollinators[25]. The 30 (out of ca. 300) Merianieae species included in this study represent two independent shifts from buzz-bee pollination to a mixed-vertebrate pollination syndrome and two independent shifts to a passerine pollination syndrome, respectively. This setup allows us to evaluate whether developmental modules persist in Merianieae flowers or whether these flowers were shaped by functional adaptation to different pollinator selection regimes and converge into distinct areas of multivariate trait space[21]. Second, all Merianieae have tubular anthers that are characteristic for a functionally highly specialized pollination system: buzz-pollination[25–27]. Only animals (i.e. bees) capable of producing high-frequency vibrations ("buzzes") can extract pollen from the small, porate openings of these anthers[27]. Although vertebrates are not capable of producing such vibrations, the tubular anther structure was retained with pollinator shifts in Merianieae and these species evolved complex alternative mechanisms of pollen expulsion[25,28]. Hence, the tubular anthers of buzz-pollinated flowers represent a text book example of a structural constraint, which apparently could not be simply reversed to 'normal' (longitudinal) anther dehiscence[25]. We can thus evaluate the role of modularity in shifting away from a functionally highly specialized and structurally constrained pollination system. Third, modularity has been linked to increased evolutionary flexibility (evolvability) and evolutionary success[6,11]. Buzz-pollination, which dominates in several large angiosperm clades, has also been associated with high evolutionary success ('adaptive plateau'[26,27]). In Merianieae, we can investigate whether modularity indeed is stronger in the buzz-bee syndrome and may be a possible explanation for the evolutionary success of buzz-pollination.

We test five alternative hypotheses of modularity in flowers (refs. [3,12], Fig. 1, Supplementary Table 1). Hypothesis 1 (developmental modularity) proposes that flowers are structured by development rather than by pollinator-mediated selection (Fig. 1[18,20,29]) and we hence do not expect to find differences in developmental modularity between the different Merianieae pollination syndromes. Hypothesis 1 is based on the different organ types which make up a flower, i.e., sterile perianth organs, male organs (stamens), female organs (carpels). Hypothesis 2 (functional modularity) is derived from the literature[12,13] and assumes flowers to be structured into a sterile pollinator 'attraction' module (sterile perianth organs, summarized as corolla here) and a fertile 'reproduction' module (stamens and carpels, Fig. 1). Since the colourful corollas are involved in visual pollinator attraction in all three pollination syndromes in Merianieae, we expect to find similar modularity in all syndromes if Hypothesis 2 is a good explanation of floral modularity. Hypotheses 3 and 4 are also derived from the literature[3,30–32] and propose functional modularity through an 'attraction' module (pollinator attraction) and an 'efficiency' module involved in mediating fit with the pollinator. Hypothesis 3 proposes that both the corolla and the conspicuous stamen appendages of Merianieae function in pollinator attraction and the stamen pore/stigma complex in efficient pollen transfer. We expect to find modularity in the buzz-bee syndrome where corollas function as landing platform and stamen appendages as handles for buzzing[25], but not in the other syndromes. Hypothesis 4 only partitions stamen appendages into the 'attraction' module while the corolla is involved in the 'efficiency' module. We expect corolla shape to be important in mediating fit with the relatively large vertebrate pollinators and hence expect to find significant modularity in the two shifted syndromes only. In addition, we put forward a functional hypothesis specific for trait functioning in Merianieae (Hypothesis 5 in Fig. 1[25]). Hypothesis 5 predicts relative functional independence of a corolla module, a stamen appendage module (functioning in pollen expulsion) and an 'efficiency' module for pollen deposition and pick-up (stamen pore/stigma complex). We expect this hypothesis to be significant in the buzz-bee and the passerine syndrome only where stamen appendages function as triggers for pollen expulsion. Finally, the comparison of each modularity hypothesis across the phylogeny of Merianieae allows us to assess the impact of evolutionary history and changes of modularity through time[18].

We find that pollinators selected for functional floral modules across developmental units and that species under the same pollinator selection regime converge in floral shape space. The ancestral bee-pollination system shows strongest floral modularity and among all floral parts, corolla shape has evolved fastest. We conclude that the strong ancestral modularity has allowed Merianieae to overcome the structural constraint of tubular anther dehiscence through increased rates of evolution in other flower parts and to flexibly adapt to changes in pollinator selection regimes.

## Results

**Floral modularity in the different pollination syndromes**. We found little support for developmental modularity (Hypothesis 1),

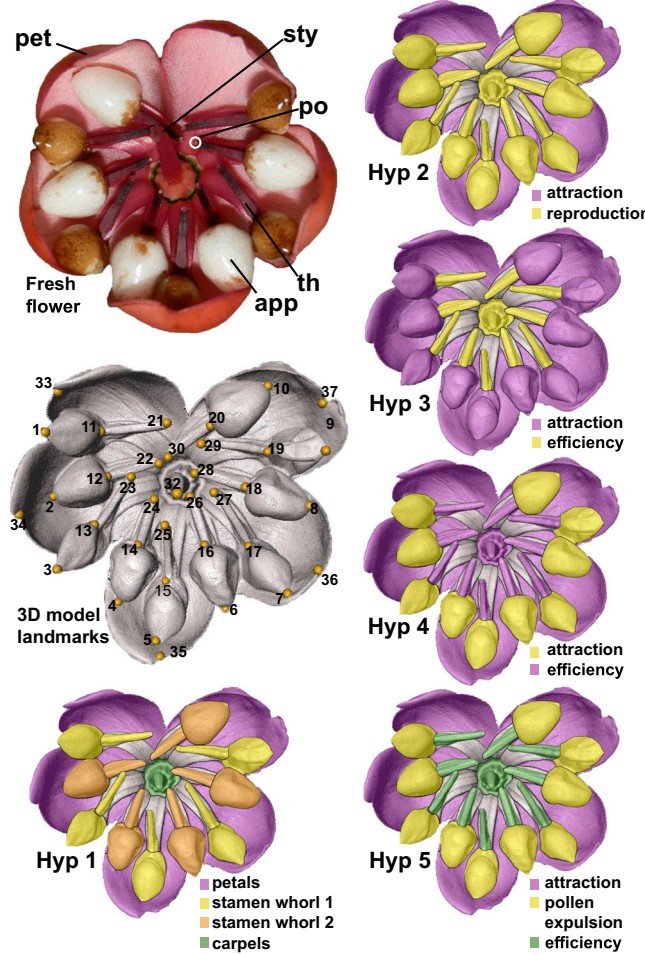

**Fig. 1 Merianieae flower, landmark configuration and the five alternative hypotheses of floral modularity, visualized on an HRX-CT scan of *Axinaea costaricensis* (passerine syndrome).** Colour patterns represent the different hypothesized modules. Example of a fresh flower: important floral structures highlighted, sterile: pet—petal; male: app—stamen appendage; th—tubular anthers (thecae) containing pollen grains; po—stamen pore from where pollen is released; female: sty—style with stigma; only ethanol-preserved flowers were used in this study. 3D model landmarks: 37 landmarks placed on 3D-model of Merianieae flowers: 1–10—stamen appendage tips, 11–20—stamen appendage base, 21–30—stamen pores, 31—base of style, 32—stigma, 33–37—petal tips. Hypothesis 1: developmental modules—four organ whorls including the sterile petal whorl, the two stamen whorls (male organs; whorl 1 and 2), and the carpel whorl (female); the sepal whorl is not landmarked as it is not involved in pollination in Merianieae. Hypothesis 2: attraction module (showy, sterile petals) and reproduction module (male and female organs[12,13]). Hypothesis 3: attraction module (showy petals and stamen appendages) and efficiency module (for pollen transfer, pore/stigma complex[31]). Hypothesis 4: alternative configuration of attraction module (colourful stamen appendages only) and efficiency module (petals, possibly also involved in mediating fit with the pollinator, and pore/stigma complex[31]). Hypothesis 5: Merianieae specific modules, attraction module (showy petals), pollen expulsion module (stamen appendages; function as handles for applying buzzes in the buzz-bee and as bellows organs for pollen expulsion in the passerine syndrome, but have lost their function in the mixed-vertebrate syndrome[25,28]), and efficiency module (pore/stigma complex for pollen transfer).

but detected significant differences in the strength of functional floral modularity (Hypotheses 2–5) among species belonging to the three different pollination syndromes in Merianieae (Fig. 2, Table 1). Flowers within the ancestral buzz-bee syndrome were

overall highly modular and the only ones to show significant modularity both in developmental and functional hypotheses. Flowers of the mixed-vertebrate syndrome showed lowest modularity, and following our expectation, the functional Hypothesis 4 was significant (Table 1). For flowers of the passerine syndrome, our analyses identified significant functional modularity as suggested by Hypotheses 4 and 5 (Fig. 2, Table 1). When comparing the strengths of modularity of the different hypotheses using effect sizes (z-scores[33]), we found effect sizes to be highest for the functional attraction/efficiency Hypothesis 4 (Table 1) in all syndromes. Overall, the strength of modularity was significantly higher in the buzz-bee and the passerine syndrome than in the mixed-vertebrate syndrome (Supplementary Table 2). The Merianieae-specific functional Hypothesis 5 had second highest effect sizes.

Although we made an effort to include as many specimens per species as possible while providing a broad sample across the Merianieae phylogeny, our sampling is still limited to only approximately 10% of Merianieae and 50% of species represented by a single specimen. Since our study relies on undamaged ethanol-preserved floral material from species which grow in remote tropical forests across Latin America, we were unable to attain more material i.e. from herbarium vouchers. To prove the robustness of our results in the light of this limited sample size, we ran extensive additional analyses using two approaches. First, we randomly rarefied our dataset 100 times to one specimen per species to understand how within-species variation may affect our results. Second, we randomly down sampled our dataset 100 times to only include 50% of all species of each pollination syndrome to understand possible bias arising from limited sampling across Merianieae.

In congruence with the results from the entire data set presented above, the buzz-bee syndrome was most modular and the only one to also show significant developmental modularity also in the additional analyses, while the mixed-vertebrate syndrome showed lowest modularity (Fig. 2, Supplementary Tables 3, 4, 5, 6). The buzz-bee and the passerine syndrome were significantly more modular in Hypothesis 4 than the mixed-vertebrate syndrome in 90% of cases (rarefaction) and 77% or 83% of cases, respectively (down sampling, Supplementary Table 6).

We assessed model fit (EMMLi[34]) in order to understand which of the five modularity hypotheses fits the data best. An additional null hypothesis (no modularity) was included in the test. In all three syndromes, the functional Hypothesis 4, partitioning the flower into an attraction (stamen appendages) and efficiency module (corolla shape, pore/stigma complex) resulted as best fit (buzz-bee AICc −1312.7, posterior probability of Hypothesis 4 74%; mixed-vertebrate AICc −801.7, posterior probability of Hypothesis 4 47 %; passerine AICc −591.4, posterior probability of Hypothesis 4 68%; Supplementary Table 7). Hypothesis 5, partitioning the flower into three functional modules, resulted as second best fit (Supplementary Table 7). When rarefying the dataset 100 times, Hypothesis 4 resulted as best fit 88% of times in the buzz-bee, 54% of times in the mixed-vertebrate and 100% of times in the passerine syndrome (Supplementary Table 8). Hypothesis 4 was also resolved as best fit for the buzz-bee and the passerine syndrome when down sampling the dataset to 50% and second highest fit for the mixed-vertebrate syndrome (Supplementary Table 8).

**Floral modularity across Merianieae.** In order to evaluate the relative evolutionary independence of floral modules, we tested the five modularity hypotheses (Fig. 1) across a molecular phylogeny of the 30 species included in this study. We found the

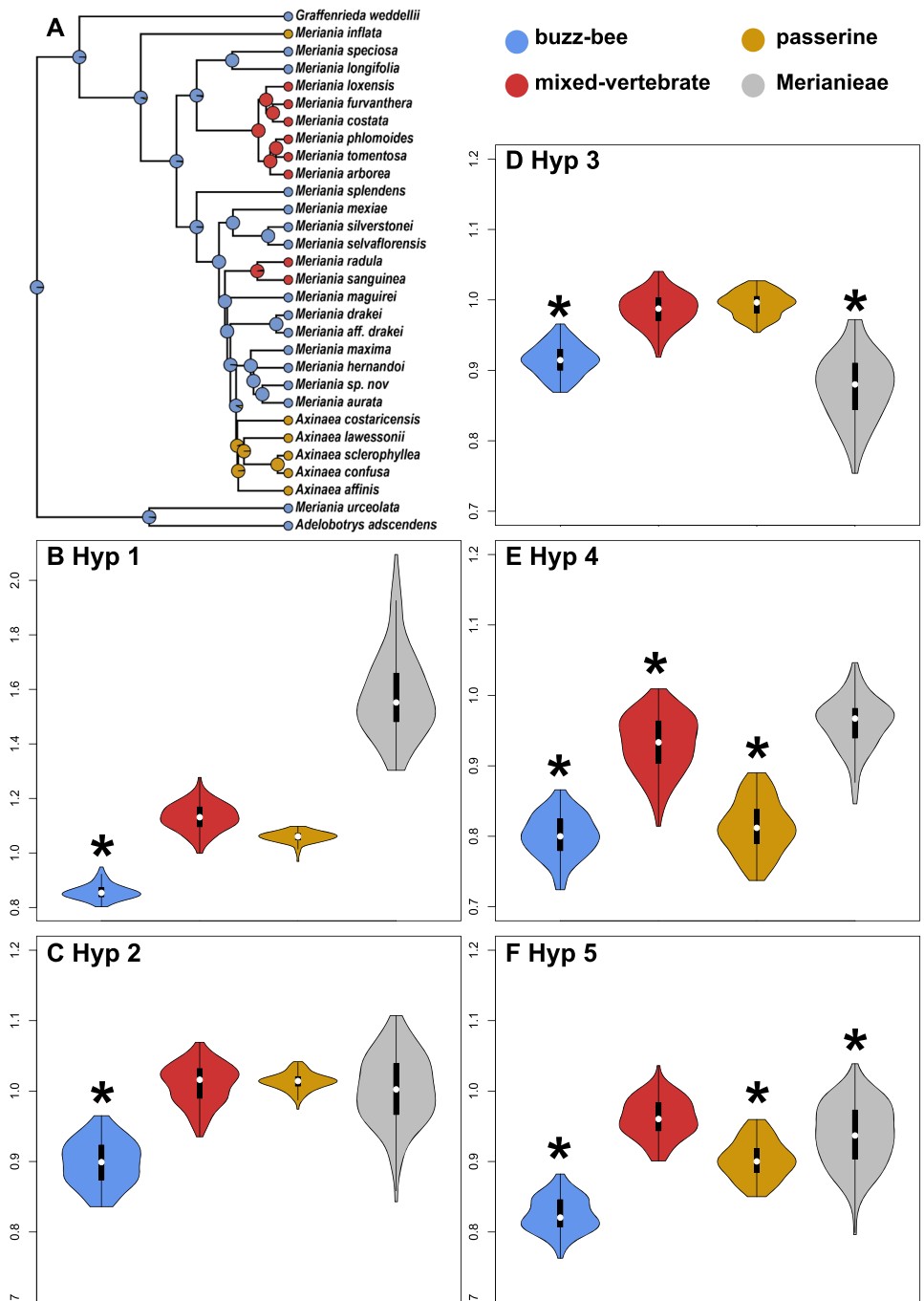

**Fig. 2 Ancestral state reconstruction of Merianieae pollination syndromes and CR-coefficients of modularity tests using rarefaction analyses. a** Buzz-bee pollination is ancestral in Merianieae and in the 30 species included in this study, two independent shifts into a mixed-vertebrate syndrome and two shifts into a passerine syndrome were detected. **b–f** CR-coefficients of modularity tests on the rarefied datasets are summarized by violin plots, medians are given as white dots, interquartile ranges as black boxes and upper/lower adjacent values as black lines; $n = 100$. Buzz-bee species always had lowest CR-values (indicating modularity). There was no modularity in mixed-vertebrate and passerine species in Hypothesis 1, Hypothesis 2 and Hypothesis 3 (**b–d**), but passerine syndrome species were modular in Hypothesis 4 and Hypothesis 5. Across Merianieae, modularity was found in more than 50% of cases in Hypothesis 3 and Hypothesis 5. Note that across Merianieae, highest CR-values were found for Hypothesis 1 (scale on y-axis up to 2.0), indicating no developmental modularity. * indicates that significant modularity was detected more than 50% of times over 100 rarefaction analyses (Supplementary Table 5).

functional Hypothesis 5 to fit the data best both in the and in the rarefied datasets (Supplementary Tables 7, 8). In the full dataset, however, no hypothesis of modularity was significant across all 30 Merianieae species (S3). The functional Hypothesis 3 and Hypothesis 5, however, were significant in more than 50% of cases in the rarefied dataset (Supplementary Table 5).

Since significant functional modularity was detected in the buzz-bee and passerine syndrome and to some extent across Merianieae (Table 1, Supplementary Table 5), we assessed whether different functional modules could evolve at different rates of morphological evolution by calculating the net rate of shape evolution of each module under Brownian motion[24].

**Table 1 Results from the five different hypotheses on modularity (Fig. 1) for the three pollination syndromes.**

| Modularity hypothesis | Buzz-bee ($n = 16$) | | | Mixed-vertebrate ($n = 8$) | | | Passerine ($n = 6$) | | | Merianieae | |
|---|---|---|---|---|---|---|---|---|---|---|---|
| | CR | p | Z | CR | p | Z | CR | p | Z | CR | p |
| Hypothesis 1 | 0.815 | *0.001* | 2.353 | 1.100 | 0.653 | 0.286 | 1.077 | 0.318 | 0.287 | 1.527 | 1.000 |
| Hypothesis 2 | 0.858 | *0.026* | 2.069 | 1.011 | 0.319 | 0.506 | 1.025 | 0.192 | 0.499 | 0.993 | 0.25 |
| Hypothesis 3 | 0.935 | 0.051 | 1.889 | 0.994 | 0.203 | 0.767 | 1.012 | 0.101 | 1.117 | 0.963 | 0.113 |
| Hypothesis 4 | 0.787 | *0.001* | 5.727 | 0.947 | *0.036* | 2.196 | 0.831 | *0.001* | 13.172 | 1.020 | 0.345 |
| Hypothesis 5 | 0.812 | *0.001* | 3.423 | 0.977 | 0.070 | 1.579 | 0.917 | *0.005* | 4.270 | 0.977 | 0.124 |

Highest degrees of modularity are present in the buzz-bee syndrome and lowest in the mixed-vertebrate syndrome, analyses of evolutionary modularity accounting for phylogenetic relatedness (column Merianieae) show significant modularity in Hypotheses 3, 4 and 5
p – p-value < 0.05 (in italics and bold) indicates significantly smaller CR than expected when no modularity is present
CR covariance ratio, Z effect sizes of CR

Indeed, the corolla/pore/stigma complex (Hyp 4, efficiency module) evolved significantly faster (sigma $4.11 \times 10^{-4}$) than the stamen appendages (sigma $1.28 \times 10^{-4}$) under Brownian motion (Hypothesis 4: $R = 3.21$, $p = 0.001$). When treating the corolla as separate module (Hypothesis 5), corolla shape evolved at least twice as fast, and significantly faster, than the rest of the flower (Hypothesis 5: $R = 4.74$, $p = 0.001$; corolla shape: sigma $6.07 \times 10^{-4}$; pore/stigma complex sigma: $3.04 \times 10^{-4}$; stamen appendages sigma: $1.28 \times 10^{-4}$). These patterns were confirmed through rarefaction analyses where corolla shape evolved fastest in 100% of cases (Supplementary Table 9).

**Flower shape evolution in Merianieae.** To test whether pollinator shifts resulted in distinct convergent floral shapes, a basic assumption of the pollination syndrome concept[21], we evaluated 3-dimensional shape evolution in Merianieae. Species have shifted repeatedly into distinct areas of morphospace and species with the same pollination syndrome indeed converged in shape (Fig. 3, variation explained: PC1 34.3%, PC2 17.9%; Supplementary Movie 1). PC1 separates the buzz-bee syndrome from the two other syndromes and captured differences in corolla shape (buzz-bee: reflexed corollas; derived syndromes: pseudo-campanulate corollas; Fig. 3a–c, Supplementary Fig. 1). PC2 separates the two derived syndromes and described differences in stamen arrangement ranging from geniculate stamens with pores close to the base of the style (buzz-bee and passerine syndromes) to partly erect stamens with pores close to the stigma; Fig. 3b, Supplementary Fig. 1).

A strong phylogenetic signal in the data indicated that flowers of closely related taxa are more similar than expected by chance ($K_{mult}$ 0.505, $p = 0.001$; rarefied dataset: average $K_{mult}$ 0.415, $p = 0.001$ in 100% of cases). We used a newly developed penalized likelihood framework[35] to estimate the fit of four different models of evolution (Brownian motion (BM), Lambda, Early-burst (EB), Ornstein-Uhlenbeck (OU)) directly on the landmark data. We found the best fit with the OU model (lowest GIC, Supplementary Table 10), which assumes evolution towards different phenotypic means as could be expected under selection mediated by different functional pollinator groups[21]. When randomly rarefying the dataset, however, the Lambda model, stretching tip branches relative to internal branches but not assuming different phenotypic optima, was resolved as best fit and the OU-model as second best fit in 100% of cases (Supplementary Table 10).

In order to test whether shifts in floral shape coincide with pollinator shifts, we estimated regime shifts on the phylogeny (L1OU[36]). As this method does not support highly multivariate landmark data, we estimated regime shifts on PC1 and PC2, respectively. We found support for four independent shifts, three of which coincide with pollinator shifts (Fig. 4, Supplementary

Fig. 2). The two buzz-bee syndrome species which also showed regime shifts have salverform corollas which are similar to the corollas found in the passerine syndrome. Our rarefaction analyses showed that these two species only shifted regimes in 35% of cases, however (Supplementary Table 11). There was no significant shift along the branch leading to *M. inflata* (passerine syndrome) or along any of the other clades with buzz-bee syndrome species. These results were supported when randomly rarefying the data. All species which have shifted pollination syndrome, except *M. inflata*, showed regime shifts in more than 50% of cases (Supplementary Table 11). The model allowing for convergence in these shifts had the best fit both in the original and the rarefed datasets (pBIC 'shifts-model' −31.4, pBIC 'convergence-model' −41.2, Supplementary Table 12).

## Discussion

Our assessment of five alternative hypotheses of floral modularity, based on 3D-models of flowers, breaks new ground in the study of floral shape evolution. We demonstrate that flowers of Merianieae are composed of modules shaped by function rather than development. This finding is well in line with a recent meta-analysis[12], which, for plants, showed that function is identified as the source of modularity more than twice as often as is development (in ca. 38% vs ca. 15% of reviewed studies[15,31], also see ref. [29]). This is in contrast to what the same meta-analysis found for animals, where modularity is explained equally often by development as it is by function (ca. 28% of studies in each case). We hypothesize that the complexity of functions performed by flowers (i.e. pollinator attraction and orientation, pollen deposition and pick-up) may be the source of such strong functional modularity in flowers. This hypothesis merits further investigations in other angiosperm lineages, for example through the comparison of floral modularity between asexually reproducing or selfing species and species which rely on cross-pollination by animals[37].

We show that pollinator-mediated selection can alter patterns and strength of modularity in flowers. In the following, we discuss how the same module may be associated with different functions in different pollination systems. In Merianieae, the corolla has undergone major changes in shape and function (summarized by PC1, convergence into pollination syndromes, Fig. 3d). In most buzz-bee syndrome species, corollas are widely open and form bowl-shaped flowers while they are more closed and form urceolate to pseudo-campanulate flowers in vertebrate pollinated species (Fig. 3). What is the functional explanation of this shape change? In all Merianieae pollination syndromes, corollas are colourful and function in pollinator attraction. In many buzz-bee syndrome species, they additionally serve as landing platforms for bees. This landing-platform function was lost with shifts to much larger vertebrate pollinators, which do not land on flowers (ref. [25],

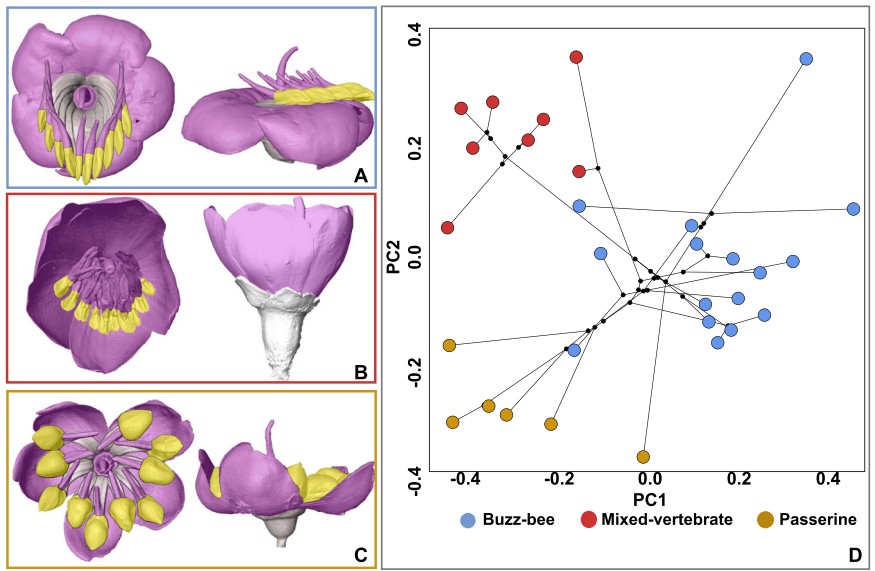

**Fig. 3 Flower shape and best-fit modularity hypothesis 4 for each pollination syndrome and phylo-morphospace on PC1 and PC2. a** Buzz-bee syndrome flower of *Meriania hernandoi*. **b** Mixed-vertebrate syndrome flower of *M. tomentosa*; there was weak support for significant modularity in this syndrome. **c** Passerine syndrome flower of *Axinaea costaricensis*. **d** PCA of mean flower shape of 30 Merianieae species with species from each pollination syndrome converging in different areas of shape space. The largest area of shape space is occupied by the buzz-bee syndrome; variation explained: PC1 34.3%, PC2 17.9%; *n* = 137 specimens.

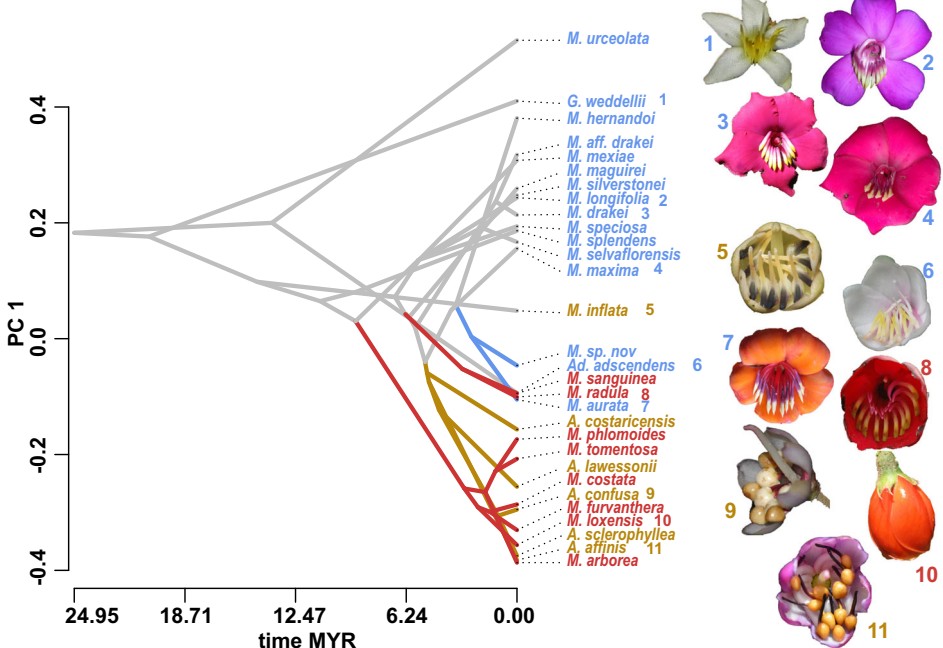

**Fig. 4 Traitgram showing floral shape evolution as summarized by PC1 (34.3% of variation explained).** The four coloured lineages show significant shifts in floral phenotypic optima as estimated by OU-models; grey branches indicate lineages that remained within the same phenotypic optimum (adaptive plateau). Note that regime shifts for two buzz-bee species (*Meriania aurata, M. sp. nov*) only occurred in 35% of rarefaction cases, while all vertebrate pollinated species shifted regime more than 50% of times. Flowers of extant taxa exemplify Merianieae floral diversity: (1) *Graffenrieda weddellii*, (2) *Meriania longifolia*, (3) *M. drakei*, (4) *M. maxima*, (5) *M. inflata*, (6) *Adelobotrys adscendens*, (7) *M. aurata*, (8) *M. radula*, (9) *Axinaea confusa*, (10) *M. loxensis*, (11) *A. affinis*.

see ref.[15] for a similar case in *Schizanthus*). We propose that the corolla has acquired a novel 'efficiency' function in these vertebrate pollination systems by restricting directions of access to the flower (Fig. 3). Vertebrate pollinators insert their tongues, bills and heads into the flowers to drink nectar or consume food body rewards (stamen appendages[25,28]). Restricting directions of access through narrower corollas may help to optimize fit with the pollinators and guarantee efficient pollen transfer[38]. Hence, the repeated evolution of pseudo-tubular corollas may be seen as a derived 'efficiency' module in Merianieae.

Our own field observations indicate that efficiency is optimized differently in the buzz-bee syndrome[25]. In these flowers, the pollen reward is contained inside the tubular stamens, and these are usually aggregated on one side of the flower, rendering flowers monosymmetric (Fig. 3a; SI Methods). Bees arrange their bodies along the stamens to extract pollen by vibration (buzzing) and the anther openings (pores) are positioned close to the stigma[25]. The monosymmetry of the androecium, therefore, likely represents the ancestral 'efficiency' function and is conserved by strong stabilizing selection to optimize mechanical fit with the bees (lower evolutionary rate[11,32,39]). Monosymmetry, albeit weaker, is also seen in the vertebrate syndromes (Fig. 3b, c). Since the vertebrate pollinators do not arrange their bodies along the stamens, however, mechanical fit with these much larger pollinators could no longer be mediated by the monosymmetry of the androecium alone, hence the additional 'derived' efficiency function of the corolla.

Buzz-bee and passerine syndrome species did not differ significantly in strength of modularity in Hypothesis 4 while modularity was much weaker in the mixed-vertebrate syndrome (Supplementary Table 2). Again, this difference may be related directly to trait functioning. In both the buzz-bee and the passerine syndrome, pollinators direct their foraging activity to the stamen appendages to obtain rewards[25]. In the mixed-vertebrate syndrome, however, stamen appendages have lost their rewarding function since nectar is secreted from the stamen filaments and aggregates on the corolla[25].

Our inability to detect significant modularity across Merianieae (Table 1) is in line with studies arguing that floral integration and modularity is likely too complex to consistently partition floral traits into the same functional modules across larger clades[39–41]. It will be interesting, however, to see whether more general patterns will arise once more studies on the modularity of complex floral architectures are available. For example, it may very well be that most flowers exhibit an 'efficiency' module (mechanical fit with the pollinator), but that these modules are constructed by different floral parts in different species or clades[3].

Theory suggests that modularity increases evolvability in organisms through reduced pleiotropic constraints[5,6,8,10,11,18]. This idea is supported by the differences in evolutionary rates that we found for two (Hypothesis 4) or three (Hypothesis 5) floral functional modules. Corolla shape evolved at a significantly higher rate (double to sixfold) than the other module(s), which is particularly important in the light of pollinator shifts and the potential to adapt to novel selection pressures[11]. 'Attraction' traits (e.g. corolla display and reward) are presumably the most important 'filters' for acquiring novel pollinators[42]. Such traits have been hypothesized to change first and more easily during pollinator shifts, followed by 'efficiency' traits, which are more conserved by stabilizing selection[11,32,42]. In accordance with these ideas, in Merianieae, corolla shape and reward type were possibly among the first traits to change during pollinator shifts[25]. The corolla acquired the derived 'efficiency' function outlined above, while the ancestral 'efficiency' module (stamen appendage position and the pore/stigma complex) apparently was more conserved and changed at a slower rate.

We hypothesize that the strong floral modularity in the ancestral buzz-bee syndrome may explain both pollinator shifts and the maintenance of the evolutionarily successful buzz-pollination system in Merianieae. First, the strong modularity in flowers of the ancestral pollination syndrome (buzz-bee) may have facilitated shifts in floral phenotype in response to major changes in selection regimes by pollinator shifts (Fig. 3[42]). Second, this strong modularity may also have enabled buzz-bee syndrome species to diversify and adapt to minor changes in selection regimes through small modular changes in the flower.

Thus, these species could explore different areas of what now appears as an 'adaptive plateau' while remaining within the buzz-bee pollinator selection regime (compared to 'adaptive wandering' by ref. [42]). This idea is supported by the buzz-bee syndrome being significantly modular in all hypotheses tested and generally more modular than the shifted syndromes (Supplementary Table 2). Testing whether maintenance of such 'adaptive plateaus' in angiosperms is facilitated by strong floral modularity, allowing for considerable flexibility to accommodate changeable environmental conditions[11], or whether it is the result of stabilizing selection conserving floral integration patterns[43], provides a fruitful challenge for future investigations.

Taking the idea of increased evolvability ahead, modularity may also have been an important pre-condition for Merianieae to overcome what we identified as a structural constraint in the tribe[25], i.e., the tubular anther structure. Such functionally and structurally highly specialized stamens with porate anther dehiscence are characteristic for pollen rewarding, buzz-pollinated flowers and have evolved multiple times independently across angiosperms[27]. While buzz-stamens possibly explain the evolutionary success of some lineages[26], they may become evolutionary dead ends when pollination by buzzing bees involves strong fitness costs (e.g. under wet, windy, cold climatic conditions in mountainous areas[44]). Only few shifts from buzz-pollination to vertebrate pollination have been documented[25,45], and we hypothesize that the reversal from the buzz-stamen to a 'normal' stamen type with longitudinal dehiscence (via a functional endothecium) is difficult (but see ref. [46]). Retaining buzz-stamens while shifting to vertebrate pollination, however, makes the evolution of new pollen-expulsion mechanisms necessary as vertebrates cannot buzz flowers. To overcome the structural constraint of the tubular anther structure, evolution apparently worked along two 'lines-of-least-resistance' in Merianieae[47–49]. The first entails modifications of the pollen expulsion mechanism from buzzing. In the mixed-vertebrate syndrome, a 'salt-shaker' pollen release mechanism has evolved. Pollen release is triggered easily when pollinators touch the anthers when inserting their mouthparts into the flowers to forage for nectar[25]. In the passerine syndrome, a highly complex 'bellows' mechanism has evolved, which is activated by foraging birds when they seize the bulbous stamen appendages with their bills for consumption[28]. As a second 'line-of-least-resistance', we propose floral functional modularity, which allowed for the independent and accelerated change of corolla shape and stamen pore position to optimize fit with the different pollinators (see above).

At a more general level, we hypothesize that strong organismal modularity may function as an evolutionary safeguard in highly specialized systems (such as buzz-pollinated flowers) by allowing lineages to evolve around structural or functional constraints[50]. Modularity in the mammalian vertebral column, for example, has been found to weaken structural constraints and may have contributed substantially to the diversity of modern mammals[51]. Experimental and comparative investigations in plants are particularly needed in order to understand the importance of modularity in facilitating adaptation to different pollinators and generating morphological diversity or stasis.

Finally, we want to mention that we are well aware that our study is based on relatively low sample sizes, including 10% of Merianieae and 50% of species only being represented by a single specimen. Geometric morphometric datasets often suffer from problems associated with small sample sizes (because of time-consuming data acquisition or little available material) but a high number of variables[24]. We aimed at minimizing such problems through choosing metrics appropriate for small and variable samples sizes (such as the CR-coefficient[24]) and verifying our results through rarefaction and down sampling analyses. As the

rarefaction and down sampling analyses corroborated our findings that modularity is significantly stronger in the buzz-bee syndrome and functional modularity characterized floral modularity better than developmental modularity, we regard our results as robust.

In conclusion, our study illustrates a novel approach to studying floral evolution by assessing the entire 3-dimensional floral architecture and testing competing hypotheses of modularity at a macroevolutionary scale. We demonstrate that pollinator-mediated selection can affect both patterns and strength of floral modularity, depending on how the different floral organs interact with pollinators. Like body parts of animals, floral modules can evolve at different rates and, in addition, modularity likely increases evolvability and may help to overcome structural constraints, thereby contributing to the striking diversity of flowers on Earth.

## Methods

**Taxon sampling and pollination syndrome classification**. Ethanol-preserved flowers of 30 Merianieae species, covering the major clades and morphological diversity of the tribe[25], were used for this study (Supplementary Table 13). Our material stems from six different Latin American countries and has been collected on various sampling trips between 2002 and 2015. Due to difficulties associated with fieldwork (research permits, species occurrence in remote and isolated places), we were unable to increase sample sizes. Fifteen out of 30 species were only represented by a single specimen, the other 15 species were represented by eight specimens on average (Supplementary Table 13). Only fully anthetic and relatively undamaged flowers were used in our study (see paragraph on Estimation of missing landmarks). For 14 species, pollinators are documented and include bees (seven sp.), passerines (three sp.) and mixed assemblages of hummingbirds, bats, rodents and flowerpiercers (five sp.)[25]. For the 16 species with unknown pollinators, the syndrome classification of Dellinger et al.[25], based on an extensive dataset of 61 floral traits not included in this study, was used. As none of the traits used for the delimitation of syndromes was used in this study, we avoid problems of circularity. Also, syndrome classification of[25] was based on rigorous field studies and objective statistical classification methods which yielded highly precise syndrome predictions. Hence, we are convinced that the risk of misclassification of species in this study is very low. In total, pollination syndromes are represented by 16 buzz-bee, eight mixed-vertebrate, and six passerine syndrome species in this study. All species have tubular anthers, releasing pollen only by a small apical pore[25]. Marked differences in pollen expulsion mechanisms differentiate the three pollination syndromes[25,28]. Stamen appendages are the key for activating pollen expulsion in the buzz-bee and passerine syndrome, while they have lost their function in the mixed-vertebrate syndrome[25].

**Phylogeny, dating and estimation of ancestral pollination syndromes**. To analyse floral shape evolution across Merianieae, we inferred a Bayesian phylogeny for Merianieae using BEAST2 (v2.5.0)[52], as implemented through the CIPRES portal[53]. We determined the best partition scheme with PartitionFinder 2[54], using each locus as a separate probable partition, and in the case of the three coding genes, also allowing for each of the three codon positions to be considered a partition. A seven partition scheme was found to be the best fit for the data (each locus as an independent partition, and in the case of ndhF, first codon position separate from second and third position). We assigned each partition the GTR + Γ + i model of sequence evolution and unlinked the partitions. We set rate variation across branches as uncorrelated and log-normally distributed, and with tree prior set to the Yule process. Based on previous analyses across the Melastomataceae, calibrated with fossils across the Myrtales, we fixed the age of the Merianieae at 29.25 MY (Michelangeli et al., unpublished). We ran three independent analyses of 60 million generations each, sampling every 20,000 generations with a 20 % burn-in. Convergence was assessed using Tracer v.1.6[55], and runs were considered satisfactory with effective sample size (ESS) values greater than 200. We combined the stable posterior distributions of the independent runs using Log-Combiner v2.5.0[56] and a maximum clade credibility tree summarized with TreeAnnotator v2.5.0[57].

We then pruned this tree to only include the 30 species present in this study (*drop.tips*; PHYTOOLS[58]). We reconstructed pollination syndromes using ML methods (*ace*; APE;[59] Supplementary Table 14) and stochastic character mapping to show that bee-pollination is ancestral (*make.simmap*; PHYTOOLS; Fig. 2a) using the 'equal-rates' model (lower AIC than 'all-rates-different'). Reconstructions of pollination syndromes were later used to paint branches using OU-models (see Flower shape evolution).

**High-Resolution X-ray Computed Tomography (HRX-CT), 3D-models**. We prepared 137 ethanol-preserved flowers of 30 species (one to 29 flowers per species, four on average; Supplementary Table 13 for exact numbers of specimens per species) for HRX-CT scanning by putting them into a contrasting agent for four weeks (1% PTA–70% EtOH, Supplementary Tables 13, 23). We then mounted fully contrasted flowers in plastic cups (Semadeni Plastics Group) and stabilized them by acrylic-pillow foam to prevent movement during the scanning process. We HRX-CT scanned the samples using the Xradia MicroXCT-200 system. We reconstructed three-D image stacks from the raw scan data (XMReconstructor XRadia Inc.) and deposited tiff-stacks on the public repository https://phaidra.univie.ac.at/ from where they can be downloaded.

**Landmark placement**. We used the imaging software AMIRA 5.5.0 to create 3D-models of the image stacks. We calculated isosurface models of each flower to place landmarks on. In total, we selected 37 landmarks under the criteria of homology and repeatability (ability to accurately locate homologous landmark positions in different specimens) to capture patterns of floral shape variation in the three different pollination syndromes (Fig. 1). We placed landmarks as follows: five on the typical notch on the petal tips, one at the base of the style (on top of the syncarpous ovary, not visible in Fig. 1), ten on the stamen appendage tips, ten on the base of the stamen appendages, ten on the anther pores, and one on the stigma. All landmarks were placed by one of us (S. A.) in order to minimize variation due to potential observer inconsistencies.

**Statistics and reproducibility**. Assessment of landmark quality. We performed all data analyses in R[60]. In order to assure accurate landmark placement and to minimize observer error, we performed a precision test at the beginning of the landmarking process for two specimens (one passerine and one hummingbird/bat pollinated) following the methodology adopted by ref. [61]. We landmarked ten replicates of the two specimens and 10 additional specimens stemming from different pollination syndromes and Procrustes fitted those three datasets separately. In optimal landmark configurations, error in replicated samples should be close to 0 and at least one magnitude smaller than in non-replicated samples. To calculate the error around each single landmark, we compared the mean distance of each landmark (of the 10 replicates and the 10 independent samples, respectively) to the consensus. Using *T*-and *F*-tests, we compared the mean replicate distances to the mean distances of the non-replicates at each landmark. All landmarks placed in both replication sets were significantly less variable than in the non-replicate placements both using *T*- and *F*-tests and observer errors (mean distance of landmarks to consensus) were more than one magnitude smaller in replicates than in the non-replicate set (set1-replicate: 0.00139, set2-replicate: 0.00117, non-replicate: 0.0689). Thus, selected landmarks were accurate enough to proceed with further landmark placement.

Estimation of missing landmarks. In 72 of the 137 specimens used for analyses, all landmarks could be placed accurately without problems. The remaining 65 specimens showed minor damages due to handling and transport or damage by herbivores or pollen thieves (e.g. broken tip of one petal, broken style tip, broken stamen or stamen tip chewed up by pollen robbing *Trigona* bees) so that one to maximally ten landmarks could not be placed. Most geometric morphometric analyses require the placement of exactly the same number of homologous landmarks in all specimens and are intolerant of missing data[62]. Our dataset includes a number of rare taxa collected at sites with difficult access from six different Latin American countries and excluding those from our analyses would have greatly reduced the breadth (in terms of taxonomic and morphological diversity) of our study. Since we aimed at capturing the actual 3-dimensional floral architecture of flowers, a study like ours also could not make use of flowers from herbarium specimens. We thus chose to estimate missing landmarks for the 65 specimens in questions, following methods developed by Arbour and Brown[62]. For these specimens, we estimated the missing landmarks by four different landmark estimation techniques (Bayesian PCA (BPCA), mean substitution (MS), thin-plate spline interpolation (TPS) and least-squares regression (REG)) using the R-package LOST (see ref. [61] for a thorough comparison of estimation techniques; J. Arbour provided updated R scripts to run TPS in 3D, currently not implemented in LOST). To improve estimation accuracy, we only estimated missing landmarks from specimens most similar to the specimen for which landmarks should be estimated[63]. Thus, we divided the dataset of the 72 intact specimens into six subsets for estimation (first column Supplementary Table 15). For each of the subsets, we performed a test run by randomly removing one to ten landmarks in one intact individual 50 times and estimating the missing landmarks. We Procrustes fitted each estimated set and performed a PCA. We used the function protest() from the R-package 'vegan' to compare PCA-coordinates (first two axes) of the estimated subset and the intact subset to test if the estimation procedure significantly altered relative morphospace occupation patterns. In addition, we used *T*- and *F*-tests to test for significant alteration of each landmark position between the estimated and the intact set in all 50 runs. All estimation techniques gave PCA results that were significantly correlated to the respective intact subset but the four techniques differed in the quality of single landmark estimation (Supplementary Table 16) with MS and REG performing worst. We chose TPS as method to estimate landmarks in all 65 specimens. In order to keep possible errors due to missing data small, we estimated each specimen with missing data separately with its respective subset.

Procrustes fitting and shape space calculation. We performed generalized Procrustes superimposition of landmarks in GEOMORPH[64] to remove variation in

position, orientation and size. For each species with more than one specimen present (15 species), we calculated the mean shape. For the other 15 species, which only were represented by a single specimen, we directly used the Procrustes fitted coordinates in subsequent analyses. We visualized shape space by Principal Component Analyses (PCA). In addition, we calculated phylomorphospaces using the *phylomorphospace* function in PHYTOOLS[58]. To visualize shape change along PC1 and PC2, we used wireframes based on codes from http://rgriff23.github.io/ 2017/11/10/ plotting-shape-changes-geomorph.html (last accessed 22 November 2018).

Testing hypotheses on modularity using the CR coefficient. We used the covariance ratio (CR) as a metric to test the five modularity hypotheses as it generates robust results even with small and variable sample sizes[24]. The CR-metric determines the degree of modularity between pre-defined modules (from our Hypotheses 1–5) and estimates whether they are significantly more modular than when landmarks are randomly re-assigned to modules (null-hypothesis of random trait association). The CR-coefficient ranges between 0 and positive values, smaller values indicate less covariation between partitions of data and hence modularity. We tested the five modularity hypotheses for each pollination syndrome separately but on joint Procrustes fitted landmark coordinates using the function *test. modularity* (GEOMORPH). Thousand random permutations were used to evaluate the statistical significance of the observed CR-coefficient.

Evaluating the strength of modularity within and among syndromes. Summary measures of trait correlation are sensitive to various attributes of the data and hence cannot be readily compared between different groups[24,33,65] such as, for instance, the three different pollination syndromes considered here. Adams and Collyer[33] proposed the z-score as a standardized test statistic for the rPLS (Partial Least Squares correlation coefficient) where the rPLS is scaled by its permutation-based sampling distribution (effect size of the rPLS is calculated as standard deviates for the permuted samples). Calculating the effect size of the difference between two rPLS effect sizes allows for direct comparison of the strength of morphological integration across datasets[33]. We extended this approach for the CR-coefficient using the formulas provided by Adams and Collyer[33] in order to statistically evaluate the strengths of modularity between the three different pollination syndromes. We performed two-sample tests to assess if levels of modularity differed significantly between pollination syndromes.

Assessing floral modularity across Merianieae. In order to understand if detected floral modules represent relatively independent units also in an evolutionary context, we tested the five different modularity hypotheses across the Merianieae phylogeny. We calculated the CR-coefficient for all species together while accounting for phylogenetic relatedness using the function *phylo.modularity* (GEOMORPH).

Selecting the best-fit hypothesis of floral modularity. The approaches outlined above allow for detection of modularity and an evaluation of the strength of modularity between the different pollination syndromes. However, they do not permit conclusions on which modularity hypothesis fits the data best. We thus used the maximum-likelihood approach proposed by Goswami and Finarelli[34] to assess the fit of the five competing hypotheses. First, vector congruence coefficient correlation matrices were calculated on the Procrustes fitted landmark coordinates for each pollination syndrome separately, resulting in three $37 \times 37$ element matrices[66] using the *dotcorr* function (PALEOMORPH;[67]). We then ran the function *EMMLi* (EMMLi;[34]) to detect the best fitting model for each pollination syndrome by comparing the finite-sample corrected Akaike Information Criterion (AICc). EMMLi allows for complex models with different correlation coefficients between and within hypothesized modules, so that a total of 15 different models were tested, including a model of no modularity. The same procedure was repeated for all species together to assess the best-fit modularity hypotheses across Merianieae.

Assessing the rate of morphological evolution. In order to understand whether different floral modules evolve at different rates (i.e. whether some traits respond to changes in pollinator selection regimes more quickly than others), we calculated multivariate net evolutionary rates under Brownian motion for each module of Hypothesis 4 and Hypothesis 5[24]. We used the function *compare.multi.evol.rates* (GEOMORPH).

Flower shape evolution. We calculated phylogenetic signal in flower shape on the landmark data by the $K_{mult}$ statistic, which is an extension of Blomberg's Kappa statistic and designed for multivariate data[68]. We then assessed the fit of four different evolutionary models (Brownian motion (BM), Lambda, Early Burst (EB), Ornstein-Uhlenbeck (OU)) to the landmark data using the newly developed penalized likelihood framework for highly multivariate datasets (*fit_t_pl* in RPANDA[35]). Based on the clear clustering of the three different pollination syndromes in shape space as assessed by PCA, we used PC1 and PC2 to visualize flower shape change on the phylogeny by constructing a traitgram (PHYTOOLS). We then modelled trait evolution (PC1–2) under an Ornstein-Uhlenbeck (OU) process[69] to screen for different phenotypic optima within Merianieae using the *l1ou* R-package[36]. We used a LASSO (Least Absolute Shrinkage and Selection Operator) procedure[70] to estimate shifts in phenotypic optima from the data without an a-priori definition of where regime shifts may have occurred (*estimate_shift_configuration* function, estimated shifts-model). Convergence of these shifts was evaluated using the *estimate_convergent_regimes* function (L1OU). We evaluated model fit using the phylogenetic Bayesian information criterion (pBIC) and calculated weights (*aicw* from GEIGER[71]).

Finally, we reconstructed morphospace evolution through time on PC1 and PC2 using the *evomorphospace* function (EVOMAP[72]). We did ancestral character estimation for PC1 and PC2 (ace, method REML, APE) and painted pollination syndromes onto branches according to the estimation of ancestral pollination syndromes (Fig. 2).

Assessing the robustness of the data. Since our dataset is limited in size (ca. 10% of Merianieae, 15 species only represented by one specimen), we worked towards carefully assessing the robustness of our results. First, we randomly rarefied the landmark dataset 100 times to only include one specimen per species. This rarefaction helps understand the impact of intraspecific variability (i.e. calculation of mean shape or representing each species by one specimen only). Second, we randomly down sampled the landmark dataset 100 times to 50% of species per pollination syndrome (hence, eight buzz-bee, four mixed-vertebrate and four passerine) to understand how a reduction in species numbers affects our results. Again, we only included a single specimen per species in each down sampled dataset. Note that we included four (instead of three) species in the passerine syndrome since this was the minimum number required in assessments of the best-fit modularity hypothesis.

We tested all hypotheses on modularity on these two additional datasets following the methods described above. We calculated CR- and p-values, z-scores and significant differences in strength of modularity between syndromes for each of the 100 runs. We summarized results by calculating average CR-, p- and z-scores (Supplementary Tables 3, 4) and by reporting the proportion of times a hypothesis of modularity was significant (Supplementary Tables 5, 6). Also, we assessed the best-fit modularity hypothesis for the subsampled datasets and summarized these results by counting how often a specific hypothesis resulted as best fit (Supplementary Table 8). We also used the rarefied datasets to compare rates of morphological evolution among modules (Supplementary Table 9 reporting averages) and tested which hypothesis of evolution fits the landmark data best (Supplementary Table 10). We further used rarefied datasets to estimate regime shifts under an OU-process of floral shape evolution. We summarized these results by calculating the proportion of times a species was included in a regime shift (Supplementary Table 11). Overall, both the rarefaction and the down sampling results are congruent with results obtained from the original data and support the view that the buzz-bee syndrome is most modular and that functional modularity better explains floral shape evolution in Merianieae than developmental modularity.

**Reporting summary**. Further information on research design is available in the Nature Research Reporting Summary linked to this article.

## Data availability
We have deposited tiff-stacks of the 3D-reconstructions of Merianieae flowers on the public repository https://phaidra.univie.ac.at/o:1016372 from where they can be downloaded free of charge.

## Code availability
All morphometric analyses were performed in R and scripts, they are available in a.zip file at https://phaidra.univie.ac.at/o:1043204.

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

## Acknowledgements

We thank field stations and personnel in Ecuador and Costa Rica for lodging and logistic support during sample collection (Bellavista Reserve, PN Podocarpus Cajanuma, Scientific Station San Francisco, EcoMinga Foundation, Yocotoco Foundation, Monteverde Biological Station, Tropical Station La Gamba). We thank Chris Klingenberg for methodological explanations, Jessica Arbour for providing codes for adapting LOST to accommodate 3D-data, Dean Adams for discussion on the adaptation of the z-score calculations for the CR-coefficient and Marion Chartier for valuable discussions on morphospaces. We further thank three anonymous reviewers for their helpful comments on this manuscript. This study was financed by the FWF (grant no. P 30669-B29) to ASD and JS and the NSF (grant no. DEB-1146409) to DSP and FAM, and the NSF (grant no. DEB-1343612 and DEB-0818399) to FAM.

## Author contributions

A.S.D. and J.S. conceived and designed the project, A.S.D., D.F.F., F.A.M., F.A. and M.A. collected floral material, A.S.D., S.A. and S.P. performed HRX-CT scanning and landmarking, A.S.D. ran morphometric data analyses, D.S.P. and F.A.M. provided sequence data and F.A.M. constructed the Merianieae phylogeny. A.S.D., J.S., D.F.F., F.A.M., W.S.A., D.S.P., F.A., M.A., S.P., S.A., and Y.S. contributed to writing and revising the manuscript.

## Competing interests

The authors declare no competing interests.
