## [Peer Review File · Communications Biology]

Reviewers' comments:

Reviewer #1 (Remarks to the Author):

This manuscript examines floral modularity in group of plants including buzz-pollinated and vertebrate pollinated flowers. I thought the core finding was intriguing — that modularity followed function as opposed to development. The authors discuss possible implications of this finding for evolution, although not why/how it could happen in the first place (interesting since apparently animals show the opposite pattern, line 159). I found the constraint aspect of the manuscript less compelling — all evolution works from some starting point, so when do we classify that starting point as a constraint? Overall though, I found the hypotheses well motivated, the analyses robust and the results interesting. Below I provide some feedback on areas where the authors could improve the ms and possibly streamline the analyses.

It feels to me like the first paragraph is trying to cover too much ground, and throws out lots of big (and often problematic) terms (like evolvability, modularity, adaptation, constraint, complexity) without sufficient context. One solution might be a more general opening paragraph about modularity before moving into flowers.

Given that the modularity varies across species with different pollination syndromes, I couldn't understand the motivation for doing a combined analysis with the whole clade (lines 117-121). Indeed the authors note on line 191-192 that "floral integration and modularity is likely too complex to consistently partition floral traits into the same functional modules across larger clades".

Minor comments:

Line 76: "Brought about the necessity" — watch out for teleological language

Line 78: Reversal to what? Non-buzz bee?

Line 113: I was confused by the contrasting results — why would model selection favor hyp. 4 for the mixed-vertebrate flowers when no modularity was detected? Could you provide a brief explanation for this?

Line 172: chase —> case

Lines 243-244: Incomplete sentence and also it's unclear (what are the re-arrangements and how do they 'overcome' the constraint?).

Fig. 2: the orange and yellow are hard to tell apart — maybe make yellow more of a brown? Then it might work for black and white if people print it off.

Reviewer #2 (Remarks to the Author):

This is an interesting study that tests different hypotheses of floral modularity in the context of pollination syndromes in Meranieae. The authors present a novel combination of 3D geometric morphometrics, model testing, and comparative phylogenetic methods to investigate shifts between pollination syndromes and overcoming the structural constraints of the specialized buzz-bee pollination syndrome.

However, the sample size of the dataset is too small to be robust. Over half of the species included in the study are represented by only a single flower specimen. Nearly half of the flowers were damaged and therefore landmarks needed to be estimated for those specimens to be included; although extensive analyses were run to determine the best estimate for these missing landmarks. Incorrect practices of leaving out extra floral organs while landmarking specimens in order to include specimens with 6 or 7 petals (as opposed to 5) were used. In addition, none of these caveats were discussed in the main text and were only included in the supplemental methods, which does not accurately portray the study. I understand that floral specimens were difficult to come by; nevertheless, this study needs additional samples.

I think this analysis is novel, important, and has great merit, but adjustments to the methodology and an increase in the sample size are necessary in order for this study to be published.

Specific comments

p. 3, lines 49-63 Specific information in the hypotheses may fit better at the end of the introduction after you've introduced the different pollination syndromes of Merianieae. In addition, a clearer description of each hypothesis is needed. I would state the question being answered, and then detail each module that makes up each hypothesis. This will help to orient the reader to your study more easily.

p. 4, line 69, Fig. S1 I would like to see this phylogeny as a figure in the text of the article, not the supplemental. It provides the basis for the relationships between species and the shifts in pollinator syndromes, which are key aspects of the paper and useful for interpreting subsequent figures.

p. 5, line 99 Hypothesis 3 is not significant for the passerine syndrome according to Table 1.

p. 5, line 104 References to Supplemental Tables in the text start with Table S5. Renumber to reflect order in text and refer to all Supplemental Information in the text.

p. 6, line 147 Extra space after (22

p. 7, line 157 Is this all land plants? A bit more detail about the study would be informative.

p. 7, line 168 I would change to 'Fig. 2' instead of 'Fig. 2D.' While the phylomorphospace may incorporate the changes, they are not apparent to the reader, whereas the pictures in A-C give an idea of what those changes look like.

p. 8, line 172 'chase' should be 'case'

p. 8, lines 173-174 Perhaps 'the direction of access' makes more sense than 'access directions.'

p. 8, line 175 Again, 'the direction of access' is clearer than 'access directions.'

p. 8, line 190 'Pollinator syndromes' instead of 'pollinators'

p. 8, line 191 'are' instead of 'is'

p. 9, line 215 Has this bellows pollen expulsion method been explained elsewhere? A brief description would facilitate the reader's understanding of the pollinator shifts.

p. 10, lines 242-244 Expand on each of these. Explain what the 'salt-shaker' and 'bellows' mechanisms entail and specifically state how re-arrangements of floral functional models overcome the structural constraint of the anther structure

p. 11, lines 258-259 How many flowers per species? This is important information that should be highlighted early on in the description of your methodology. The reader must be able to evaluate whether the sample size of your study is sufficient.

p. 12, lines 283-284 How was the resampling done? How many flowers were included per species in the resampling datasets?

p. 12, line 290 Include more information on the details of the covariance ratio test

Fig. 1 In the key of Hyp 5, 'efficiency' is misspelled

Fig. 2 The salmon and other colors used to delineate mixed-vertebrate and passerine pollination syndromes are much too similar to each other and it is hard to determine which samples are which. Choose a different color scheme for your figures. Also, when I printed the figures in black and white, all colors used were the same saturation of gray.

Fig. 2D Add the % variation explained by each PC to the axes.

Fig. 3 Again, the color scheme does not clearly distinguish mixed-vertebrate from passerine. The % variation explained by PC1 could also be added to the y-axis here.

Table 1 Consider adding the species sample size for each pollination syndrome in the table headings, e.g. buzz-bee (n=19)

Supplemental Methods

I was very surprised to find that there were several key factors about the methodology of the study in supplemental methods that were completely left out of the main text of the article. None of the caveats (small sample sizes, flowers with extra organs, estimation of missing landmarks, difficulty of obtaining specimens) of this study were brought up in the main text, which results in a very different impression of the robustness of the study when reading the main text versus the supplemental methods. It is not appropriate to relegate problems with a study to the supplemental methods. All aspects of the study must be explained in the main text for transparency and repeatability.

Information from Supplemental Methods 1.1, 1.3, and 1.5 must be included in the main text to accurately describe the methodology of the study. In addition, the main text should reference the specific sections of the Supplemental Methods as appropriate to ensure that the reader can easily find the pertinent supplementary information.

Table S1 About half of the species in this study are represented by a single floral specimen. This does not account for any intraspecific variation and is not a robust dataset. Morphometric analyses should have at least 5 flowers per species/group. Only one third of the species included here have 5 or more flowers.

Section 1.1 It is not appropriate to leave out the extra floral organs from hexa- and heptamerous flowers when applying landmarks. This does not accurately represent the shape of these flowers or what shape the flowers would have if they were pentamerous. The flower specimens with extra floral organs cannot be included in this dataset. How many of the flowers used had extra floral organs?

Section 1.3 Half of the specimens used needed to have landmarks estimated to be included. This is far from ideal and using other species most similar to the damaged specimens to estimate the landmark may result in a greater degree of similarity between preconceived groups than actually present. However, the authors have included extensive analyses to test whether the estimated landmarks seem accurate, making the best out of a less than ideal situation. The authors note that 1-10 landmarks were estimated. Perhaps the dataset can be limited to estimating 1 or 2 landmarks, which would limit the amount of uncertainty in the dataset. As mentioned above, the fact that landmarks were estimated for half of the specimens must be included in the main text for transparency.

Section 1.4 How were the resampled datasets sampled? How many flowers per species were included? How were these datasets used in subsequent analyses? Was the average used, or each dataset run?

Table S7 An explanation of what 'same.' and 'sep.' and '.between' mean in the table legend would be useful for interpretation.

Reviewer #3 (Remarks to the Author):

Comments for "Is modularity the key to adaptive success in buzz-pollinated flowers?" by Dr Dellinger and colleagues [Paper # COMMSBIO-19-0707-T].

In this manuscript, the authors tried to abstract the sets of traits or functions of flowers as

"modularity", tested five parallel hypotheses to see whether modularity is crucial to adaptation success in buzz-pollinated flowers. They selected 33 species from Merianieae (with ca. 300 species), constructed 3-D flower models based on alcohol-soaked materials, constructed phylogenetic trees with time-calibration, analyzed whether pollinator shift is associated with disruption of floral modularity. The conclusion of this study is that floral modularity may be the key to adaptive success for those buzz-pollinated flowers in their study system.

Overall, this study consists of great efforts in both sampling and analyzing. The methods are as the authors stated as "state-of-the-art", presenting a new approach to document the evolution of floral traits. The results sound clear and convinced. If concerns listed below can be solved, the significance of the study could be better improved.

Two major concerns are:

(1) whether the study system is a little narrow for readers' interest

To understand floral evolution, the concept of "modularity" could be of great significance and with broad interest, however, it seems unclear why the authors focused on buzz-pollinated flowers? Buzz-pollination is one of specialized pollination systems, the reason to use it as the study model is lacking, regardless the taxa that chosen as the representative of evolutionary shifts in buzz-pollination system are reasonable. It would be better to briefly claim the necessity of taking buzz-pollinated flowers as a study system if a vast scenario could be addressed.

(2) What are different between the concepts of "modularity" and "pollination syndrome"?

The term "modularity" used in this paper is quite easy to remind the concept of "pollination syndrome" for people who are interested in evolution of flowers, because both "modularity" and "pollination syndrome" depict a set of traits or function that flowering plants may possess. For instance, in L38, "modularity" was defined as "organisms may consist of several trait clusters", while "pollination syndrome" is generally to describe the feature of trait clusters. (or what is difference with trait co-variation?) It seems that this paper intentionally weakened the relationship between "modularity" and "pollination syndrome". It seems to me this is not necessary. Some comparative illustrations can help readers to tell the difference between these two concepts, and build up new links to "modularity". If not, the start point of this study seemed to be less powerful, for instance, in L43-48, the authors stated "This gap in our knowledge is surprising since flowers represent ideal systems to test hypotheses of modularity and may potentially expand the existing concepts of shape evolution", however, similar tests for the concept pollination syndrome has long been prevalent, and systematic. Try to eliminate the impression that the study is trying to elucidate the same thing in different manner.

Several minor points for modifications:

L27 If no comparison was made, "also" would be improper to be used here.

L49-63 The introduction of these hypotheses that the study was trying to test, could be listed more clearly. It is not recommended that the authors sum up the hypotheses at the very beginning before introducing what each hypothesis is. In L52, the readers do not know what exactly the five hypotheses are, while the authors quickly offered the readers with a series of sum-ups: "the four other hypotheses assume...", "and two combinations of slightly differing...". As I understand, the authors could give a list of these five hypos, and concisely state what these hypotheses would predict. Then they may give sum-ups after this list. To some extent, the current form of L49-63 is in a mess.

L64-65 I agree that Merianieae represents an excellent system to study the role of floral modularity, but a shocking fact is that, this study contained only 33 out of ca. 330 Merianieae species. Are these 33 species representative for Merianieae? Is there any criterion in selecting study species? Is there any bias in selecting these species? For example, 33 species from various clades are completely sampled.

L258-259, the difference between alcohol preserved, FAA preserved, and fresh flowers should be mentioned here. Alcohol or FAA preserved materials may be swelled, potentially resulting in different results. This statement can not be omitted, because in Fig.1, the flower is marked with "fresh flower", showing the idea of this study was based on fresh flower instead of chemically preserved flowers.

L261-263, Pollinators for 18 out of 33 species are unknown. Therefore, the power of pollination syndrome should be concerned as the authors deduced pollinator type by syndrome classification indirectly or from literature rather than field observations. This reduced the credibility of the study. The extent, to which such indirect evidence can be convinced, could be illustrated.

L270-272 The necessity of calibrating phylogenetic time is not clear. If I am not wrong, the results of the time-calibrated tree are only used to make a supplementary video at least in this study here.

L275-277. The sample size varies widely among species, and is quite small in some species. Although such a factor has been considered in the analyses, a claim of the limited samplings would be appreciated in testing these hypotheses and illustrating their ideas.

Shuang-Quan Huang with help of Dr. Tong-Ze Yu from CCNU, China.

Dear reviewers,

Thank you very much for your constructive and detailed feedback on our study on modularity and 3D-shape evolution in Merianieae. We have gone through all comments and did our best to satisfy your requests and resolve issues associated with our data, including re-analyses of all our datasets.

Before answering your questions more specifically, we respond to two points raised by both reviewer #2 and reviewer #3. First, both reviewers were worried about the limited sample size of our study, only including approximately 10% of Merianieae species and 50% of specimens only being represented by one specimen. Reviewer #2 asked for the inclusion of additional specimens, but unfortunately, we are absolutely unable to comply with this request. The specimens we are studying have been collected during multiple, time consuming field excursions to six different Latin American countries and grow in remote places that are difficult to access and require special collection permits. We made an effort to include as many specimens per species as possible while providing a broad sample across the Merianieae phylogeny. Our sampling, although limited in species number, covers the major clades of Merianieae and includes as many species as we could get good floral material of. We want to emphasize that a study like ours relies on ethanol-preserved floral material and cannot make use of dried herbarium specimens or highly damaged flowers. For the 16 species where we only have one specimen, no more floral material is available.

We understand reviewer #2's and #3's concerns about this limited sample size and associated issues with the robustness of our data. Since we could not increase sample sizes, we made great efforts to increase the transparency of our data through methodological clarifications and additional analyses. We had used a rarefaction analysis in addition to calculating modularity and shape evolution on floral mean shape (mean for species where multiple specimens were available, otherwise Procrustes fitted coordinates) in the initial submission, but believe that our results have not been presented clearly enough in the earlier version of the manuscript. Hence, we refined these sections of the manuscript and clearly state the results of our rarefaction analyses where we randomly subsampled the dataset to one specimen per species 100 times. We report these results in a new figure (Fig. 2), in the main text and in six tables in the SI (Table S3, S5, S6, S8, S10, S11; tables S5, S6, S8 are new). With this rarefaction analysis, we are able to show that the overall results of our study remain the same even when only including one specimen per species for all species. To better understand the impact of only including a limited number of specimens, we performed an additional analysis randomly down sampling the dataset to 50% of species per pollination syndrome 100 times. In this down sampling analysis, only 8 out of 16 buzz-bee syndrome species, 4 out of 8 mixed-vertebrate syndrome species and 4 out of 6 passerine syndrome species were included at random. Please note that we decided to use 4 instead of 3 passerine syndrome species since the determination of the best-fit model of modularity (EMMLi) did not allow for including fewer than four specimens. These down sampling analyses overall confirmed results that we had obtained using the mean shape and the rarefied datasets and have been reported in the main text and in the SI tables S4, S5, S6 and S8. Hence, the reader may evaluate where and to what extent our results may be biased by sample size issues. Overall, however, these additional analyses prove our point that function is more important than development in structuring floral modularity in Merianieae and that modularity overall is strongest in the ancestral buzz-bee pollination syndrome and weakest in the shifted mixed-vertebrate pollination syndrome.

The second point of criticism raised by reviewers #2 and #3 was that we had included specimens with more than five petals in our study. We had discussed this approach earlier with morphometricians and concluded that, when studying shape functioning, this approach may be valid since our

landmarking captured aspects of shape relevant for the interaction with pollinators such as the width of the androecium or the shape of the corolla. As we are also studying developmental aspects of modularity and evolution, however, we agree with reviewers #2 and #3 that we should remove these specimens from our dataset. Hence, we have removed the 10 specimens which had more than five petals. Unfortunately, this also led to a reduction in the number of species included from 33 to 30. All analyses presented in the current manuscript are now based on these 30 species only.

In the following, we will respond to the concerns of each reviewer more specifically, our responses are highlighted in *italics* writing. Some minor comments such as typos have not been commented on here but we have adjusted those directly in the text.

Reviewer #1:

Thank you very much for your positive feedback on our study.

This manuscript examines floral modularity in group of plants including buzz-pollinated and vertebrate pollinated flowers. I thought the core finding was intriguing — that modularity followed function as opposed to development. The authors discuss possible implications of this finding for evolution, although not why/how it could happen in the first place (interesting since apparently animals show the opposite pattern, line 159). I found the constraint aspect of the manuscript less compelling — all evolution works from some starting point, so when do we classify that starting point as a constraint? Overall though, I found the hypotheses well motivated, the analyses robust and the results interesting. Below I provide some feedback on areas where the authors could improve the ms and possibly streamline the analyses.

We provide a short section at the beginning of the introduction on why we think function may be more important than development in structuring modularity in flowers, but we refrain from going into detail since we believe that, at the current state of knowledge, this pattern is not understood well enough.

It feels to me like the first paragraph is trying to cover too much ground, and throws out lots of big (and often problematic) terms (like evolvability, modularity, adaptation, constraint, complexity) without sufficient context. One solution might be a more general opening paragraph about modularity before moving into flowers.

We worked at providing more context on modularity at the beginning of the introduction as you suggested.

Given that the modularity varies across species with different pollination syndromes, I couldn't understand the motivation for doing a combined analysis with the whole clade (lines 117-121). Indeed the authors note on line 191-192 that “floral integration and modularity is likely too complex to consistently partition floral traits into the same functional modules across larger clades”.

We argue that the analysis of evolutionary modularity across the clade is important to understand whether any of the modules identified within syndromes also persist across syndromes and are potentially evolutionarily stable. We think it would be wrong to a-priori assume that there are differences between species with different pollinators without actually testing it while accounting for

phylogenetic relatedness.

All minor comments have been dealt with in the text, we have also changed the colour schemes of our figures.

Reviewer #2:

However, the sample size of the dataset is too small to be robust. Over half of the species included in the study are represented by only a single flower specimen. Nearly half of the flowers were damaged and therefore landmarks needed to be estimated for those specimens to be included; although extensive analyses were run to determine the best estimate for these missing landmarks. Incorrect practices of leaving out extra floral organs while landmarking specimens in order to include specimens with 6 or 7 petals (as opposed to 5) were used. In addition, none of these caveats were discussed in the main text and were only included in the supplemental methods, which does not accurately portray the study. I understand that floral specimens were difficult to come by; nevertheless, this study needs additional samples.

As outlined above, we have done our best to assess the robustness of our limited dataset through rarefaction and down sampling analyses while being unable to increase our sample size. Also, we have revised sections of the Material and Methods to increase transparency and included details on the estimation of missing landmarks and assessments of the robustness of our data.

Since we are unable to comply with your request of including additional specimens (see above), we want to highlight why our study is of great importance to the scientific community:

1) Studies on floral adaptations to pollinators have a long tradition, but are usually limited to qualitative trait coding or simple morphometric measurements in two dimensions only, ignoring the 3D architecture of flowers. The few studies which have quantified flower shape in 3D have focused on corolla shape and hence ignored subtler shape differences, i.e. associated with the relative position of the reproductive organs. Studying the entire floral architecture in 3D is crucial for advancing our understanding of pollinator mediated selection on flowers. Our study presents a methodological advance in studying flower shape by considering all floral organ types and testing five different hypotheses on flower evolution. This study provides both information on the methodological toolkit needed to capture, quantify and analyse 3D flower shape evolution and at the same time exemplifies this approach in a group of exotic tropical plants.

2) Our sampling size issues arise from featuring tropical plant species which occur in remote, isolated places, many of which are threatened by deforestation, across South America. While we do not want to gloss over the methodological problems, we want to emphasize the importance and value of studying such non-model organisms. Merianieae are most diverse in the tropical Andes, the world's most species-rich biodiversity hotspot, yet, very little is known about the lineages which make up this diversity, let alone their evolution and natural history. Quantifying floral evolution and assessing rates of morphological evolution, convergence and adaptation allows for deepening our understanding of how this tremendous diversity may have arisen and what drove its evolution. Only increasing the number of studies which address these questions in non-model organisms will allow for drawing more general conclusions.

Specific comments

p. 3, lines 49-63 Specific information in the hypotheses may fit better at the end of the introduction after you've introduced the different pollination syndromes of Merianieae. In addition, a clearer description of each hypothesis is needed. I would state the question being answered, and then detail each module that makes up each hypothesis. This will help to orient the reader to your study more easily.

We restructured the introduction and hope that the hypotheses are presented more clearly now.

p. 4, line 69, Fig. S1 I would like to see this phylogeny as a figure in the text of the article, not the supplemental. It provides the basis for the relationships between species and the shifts in pollinator syndromes, which are key aspects of the paper and useful for interpreting subsequent figures.

A similar phylogeny representing pollinator shifts has already been published by us earlier this year (Dellinger et al. 2019, New Phytologist, <https://nph.onlinelibrary.wiley.com/doi/full/10.1111/nph.15468>) and we wanted to avoid presenting a similar result twice. We agree with you, however, that this phylogeny forms the basis for the hypotheses tested, we decided to include it in Figure 2 now.

p. 9, line 215 Has this bellows pollen expulsion method been explained elsewhere? A brief description would facilitate the reader's understanding of the pollinator shifts.

p. 10, lines 242-244 Expand on each of these. Explain what the 'salt-shaker' and 'bellows' mechanisms entail and specifically state how re-arrangements of floral functional models over comes the structural constraint of the anther structure

Since these pollen expulsion mechanisms have also been described in the above-mentioned publication and an earlier publication (Dellinger et al. 2014, Current Biology, [https://www.cell.com/current-biology/fulltext/S0960-9822\(14\)00634-4?returnURL=https%3A%2F%2Flinkinghub.elsevier.com%2Fretrieve%2Fpii%2FS0960982214006344%3Fshowall%3Dtrue](https://www.cell.com/current-biology/fulltext/S0960-9822(14)00634-4?returnURL=https%3A%2F%2Flinkinghub.elsevier.com%2Fretrieve%2Fpii%2FS0960982214006344%3Fshowall%3Dtrue)), we did not elaborate on them too much in the earlier manuscript. To provide more context in the current paper, however, we have extended this explanation.

p. 11, lines 258-259 How many flowers per species? This is important information that should be highlighted early on in the description of your methodology. The reader must be able to evaluate whether the sample size of your study is sufficient.

We included references to the SI Table S12 where all details on sample sizes can be seen.

p. 12, lines 283-284 How was the resampling done? How many flowers were included per species in the resampling datasets?

We worked at clarifying these points both in the main text and in the SI.

p. 12, line 290 Include more information on the details of the covariance ratio test

As mentioned above, we have included a brief explanation of what the test does.

Fig. 2 The salmon and ocher colors used to delineate mixed-vertebrate and passerine pollination syndromes are much too similar to each other and it is hard to determine which samples are which.

Choose a different color scheme for your figures. Also, when I printed the figures in black and white, all colors used were the same saturation of gray.

We redrew all figures with stronger colours and hope that this alleviates the problems of colours not being well distinguishable.

Fig. 2D Add the % variation explained by each PC to the axes.

We added this in the figure's description.

Table 1 Consider adding the species sample size for each pollination syndrome in the table headings, e.g. buzz-bee (n=19)

We added the number of species per syndrome.

Supplemental Methods

I was very surprised to find that there were several key factors about the methodology of the study in supplemental methods that were completely left out of the main text of the article. None of the caveats (small sample sizes, flowers with extra organs, estimation of missing landmarks, difficulty of obtaining specimens) of this study were brought up in the main text, which results in a very different impression of the robustness of the study when reading the main text versus the supplemental methods. It is not appropriate to relegate problems with a study to the supplemental methods. All aspects of the study must be explained in the main text for transparency and repeatability.

Information from Supplemental Methods 1.1, 1.3, and 1.5 must be included in the main text to accurately describe the methodology of the study. In addition, the main text should reference the specific sections of the Supplemental Methods as appropriate to ensure that the reader can easily find the pertinent supplementary information.

We worked towards referring to these sections and explaining the caveats of our study at various points in the text, and include more details on the methodology underlying our study.

Table S1 About half of the species in this study are represented by a single floral specimen. This does not account for any intraspecific variation and is not a robust dataset. Morphometric analyses should have at least 5 flowers per species/group. Only one third of the species included here have 5 or more flowers.

As highlighted above, we can unfortunately not change this fact. We want to emphasize that, in our tests on modularity, we test each hypothesis within a pollination syndrome, and each pollination syndrome is represented by more than 5 species, hence more than five specimens are included in each comparison, including the randomly rarefied analyses. Each pollination syndrome is represented by more than 30 specimens (buzz-bee 57, mixed-vertebrate 37, passerine 43).

Section 1.1 It is not appropriate to leave out the extra floral organs from hexa- and heptamerous flowers when applying landmarks. This does not accurately represent the shape of these flowers or what shape the flowers would have if they were pentamerous. The flower specimens with extra floral organs cannot be included in this dataset. How many of the flowers used had extra floral organs?

10 out of 147 specimens had additional floral organs and we removed those from our analyses, as outlined above. We had discussed this landmarking approach with various morphometricians and chose to include the most lateral stamens since they represent the functional properties of the flower (i.e. the width of the androecium is much more important in the pollination process than the actual number of stamens) and it was our aim to study functional aspects of pollination. We agree with you, however, that since we also conduct tests on developmental modularity and assess evolutionary modularity, this approach is incorrect.

Section 1.3 Half of the specimens used needed to have landmarks estimated to be included. This is far from ideal and using other species most similar to the damaged specimens to estimate the landmark may result in a greater degree of similarity between preconceived groups than actually present. However, the authors have included extensive analyses to test whether the estimated landmarks seem accurate, making the best out of a less than ideal situation. The authors note that 1-10 landmarks were estimated. Perhaps the dataset can be limited to estimating 1 or 2 landmarks, which would limit the amount of uncertainty in the dataset. As mentioned above, the fact that landmarks were estimated for half of the specimens must be included in the main text for transparency.

We included a sentence in the methods stating that we have estimated landmarks. Unfortunately, limiting estimation to 1 or 2 landmarks will not help since when a single stamen is missing, e.g. because it broke off in the process of transportation, already 3 landmarks are missing. We were very careful when estimating landmarks and as you see in the SI methods, have conducted various analyses to assure that our estimations accurately represent flower shape. We hence believe that this approach does not have an impact on our results.

Section 1.4 How were the resampled datasets sampled? How many flowers per species were included? How were these datasets used in subsequent analyses? Was the average used, or each dataset run?

In the rarefaction analyses, we randomly selected one specimen per species for those species where more than one specimen was available, for the other species, we included the single available specimen. We repeated this rarefaction 100 times and ran all tests on modularity and model fit on the rarefied datasets. Hence, we calculated CR-values, associated p- and z-values and tested for significant differences in strength of modularity between pollination syndromes on each rarefied dataset. In the revised manuscript, we present these results in a new Figure (Fig. 2) which shows violin plots of the 100 rarefaction analyses and hence represents the variability in the data. In supplementary table S3, we present averages of CR-, p-, and z-scores, in S5 we report the number of times a specific hypothesis was found to be significant and in table S6 we report the number of times strength of modularity differed significantly between pollination syndromes. We also estimated fit of the modularity hypotheses on the rarefied dataset (Table S8). In addition, we also estimated regime shifts through OU models on the rarefied datasets and counted the number of times a species was found to undergo a significant regime shift.

Finally, we also ran modularity tests on down sampled datasets during the revision process. We down sampled each pollination syndrome to only include 50% of species. Again, this was done 100 times and in each run, only one specimen per species was picked at random (if more than one specimen was available for the randomly selected species to include in each run). As with the rarefaction analyses, we calculated average CR-, p- and z-scores (table S4) and counted the number of times a specific hypothesis of modularity was found to be significant (S5) and the number of times strength of modularity differed significantly between syndromes (S6). We also estimated model fit on the down

sampled dataset (S8). We did not run analyses across Merianieae for the down sampled datasets since those datasets will likely not represent a broad sample across the phylogeny that we aimed at in our initial sampling of 30 species. If you wish to see results of down sampling across the phylogeny, however, we may run these analyses as well.

Finally, these rarefaction and down sampling analyses overall are in line with our results obtained on mean shape, stating that functional modularity was more important in Merianieae flowers than developmental modularity and that modularity is strongest in the ancestral buzz-bee syndrome. There is some disagreement about the best-fit hypotheses of modularity, and we clearly report these results in the SI tables. We hope that we could convince you of the robustness of our results, despite the small size of our dataset.

Table S7 An explanation of what 'same.' and 'sep.' and '.between' mean in the table legend would be useful for interpretation.

Included.

Reviewer #3:

Thank you for your review, we want to clarify some points and comments on your points of criticism in the following.

Two major concerns are:

(1) whether the study system is a little narrow for readers' interest

To understand floral evolution, the concept of "modularity" could be of great significance and with broad interest, however, it seems unclear why the authors focused on buzz-pollinated flowers? Buzz-pollination is one of specialized pollination systems, the reason to use it as the study model is lacking, regardless the taxa that chosen as the representative of evolutionary shifts in buzz-pollination system are reasonable. It would be better to briefly claim the necessity of taking buzz-pollinated flowers as a study system if a vast scenario could be addressed.

We agree that in the initial version of the manuscript, there was no mentioning of why buzz-pollination represents an ideal system to study in the context of pollinator mediated selection and modularity. We have included our arguments in the introduction. We want to emphasize that buzz-pollination represents a functionally and structurally highly specialized pollination system that has arisen multiple times independently across disparate angiosperm groups. Also, it has become the dominant form of pollination in some lineages such as the mega-genus Solanum (1500 species) or the large plant family Melastomataceae (ca. 5000 species). There has been considerable discussion on the role of constraint in specialized pollination systems and whether specialization leads to an evolutionary dead end. In our study, we demonstrate that even a structurally and functionally specialized system such as buzz-pollinated flowers may evolve and change, and that modularity plays a crucial role in the context of enabling adaptation to novel pollinators. We agree that there are many other systems where such hypotheses should be tested and we hope that, with our study, we stimulate further research tackling this topic in order to generate a broader understanding of the role of floral modularity in pollinator shifts.

(2) What are different between the concepts of "modularity" and "pollination syndrome"?

The term "modularity" used in this paper is quite easy to remind the concept of "pollination

syndrome” for people who are interested in evolution of flowers, because both “modularity” and “pollination syndrome” depict a set of traits or function that flowering plants may possess. For instance, in L38, “modularity” was defined as “organisms may consist of several trait clusters”, while “pollination syndrome” is generally to describe the feature of trait clusters. (or what is difference with trait co-variation?) It seems that this paper intentionally weakened the relationship between “modularity” and “pollination syndrome”. It seems to me this is not necessary. Some comparative illustrations can help readers to tell the difference between these two concepts, and build up new links to “modularity”. If not, the start point of this study seemed to be less powerful, for instance, in L43-48, the authors stated “This gap in our knowledge is surprising since flowers represent ideal systems to test hypotheses of modularity and may potentially expand the existing concepts of shape evolution”, however, similar tests for the concept pollination syndrome has long been prevalent, and systematic. Try to eliminate the impression that the study is trying to elucidate the same thing in different manner.

We disagree with this statement since we do not think that modularity addresses the same topics as pollination syndromes. As you correctly note, tests of pollination syndromes have been systematic, and we are very well aware of this since we ourselves have performed a systematic search of pollination syndromes in Merianieae, including random forest analyses and a set of 61 floral traits and field observations (Dellinger et al. 2019, New Phytologist, <https://nph.onlinelibrary.wiley.com/doi/full/10.1111/nph.15468>). The pollination syndromes we are using in this paper are based on the earlier study on pollination syndromes. In this paper, we address the concept of modularity, which, as you say, assumes that within an organism, one subset of traits may show very strong correlations within these traits, but weak correlations with another subset of traits, and hence represent a module. Modularity may have different origins, including developmental, genetic, functional and evolutionary relationships. Obviously, functional modules may differ between species which underlie different selection regimes, hence, species belonging to different pollination syndromes may show different functional modularity. But this is by no means clear, and does not preclude the existence of developmental or evolutionary modules regardless of pollination syndromes. We study floral shape across the Merianieae phylogeny and approach the study of modularity neutrally by analysing modularity both across Merianieae and within pollination syndromes, but we do not try to elucidate pollination syndromes through modularity. Pollination syndromes describe trait clusters, but they do not attempt to uncover the underlying sources of floral correlation structures. In this study, we analyse correlation structures (modules) within flowers and test whether and how correlation structures change with pollinator shifts, and this is a very different topic than pollination syndromes.

L49-63 The introduction of these hypotheses that the study was trying to test, could be listed more clearly. It is not recommended that the authors sum up the hypotheses at the very beginning before introducing what each hypothesis is. In L52, the readers do not know what exactly the five hypotheses are, while the authors quickly offered the readers with a series of sum-ups: “the four other hypotheses assume...”, “and two combinations of slightly differing...”. As I understand, the authors could give a list of these five hypos, and concisely state what these hypotheses would predict. Then they may give sum-ups after this list. To some extent, the current form of L49-63 is in a mess.

We aimed at explaining hypotheses more clearly and restructuring the introduction.

L64-65 I agree that Merianieae represents an excellent system to study the role of floral modularity,

but a shocking fact is that, this study contained only 33 out of ca. 330 Merianieae species. Are these 33 species representative for Merianieae? Is there any criterion in selecting study species? Is there any bias in selecting these species? For example, 33 species from various clades are completely sampled.

*Yes, the 30 species included in the revised version of the manuscript are representative and cover the major clades and morphological diversity of Merianieae (also compare Dellinger et al. 2019, New Phytologist, <https://nph.onlinelibrary.wiley.com/doi/full/10.1111/nph.15468>). No single clade is sampled completely. We aimed at sampling broadly across the bee-pollinated species and including also species from lineages where no pollinator shifts have occurred (e.g. *Adelobotrys adscendens*, *Graffenrieda weddelii*). Also, we aimed at representing species which have shifted pollinators by at least two independent shifts. As noted above, our sampling was constrained by the difficulty of acquiring floral material suitable for a study on 3D-flower shape where we need fully anthetic, undamaged, ethanol preserved flower material.*

L258-259, the difference between alcohol preserved, FAA preserved, and fresh flowers should be mentioned here. Alcohol or FAA preserved materials may be swelled, potentially resulting in different results. This statement can not be omitted, because in Fig.1, the flower is marked with “fresh flower”, showing the idea of this study was based on fresh flower instead of chemically preserved flowers.

The study is only based on ethanol preserved flower material so that no artefacts of tissue change through preservation should affect our results.

L261-263, Pollinators for 18 out of 33 species are unknown. Therefore, the power of pollination syndrome should be concerned as the authors deduced pollinator type by syndrome classification indirectly or from literature rather than field observations. This reduced the credibility of the study. The extent, to which such indirect evidence can be convinced, could be illustrated.

Here, we have to emphasize that syndrome delimitation has been done by ourselves and is based on extensive field observations and a detailed trait dataset of 61 floral traits, none of which is included in the present study. Hence, there are no issues of circularity in our present dataset and convergence into syndromes which we find here prove the quality of the initial syndrome delimitation (see Dellinger et al. 2019, New Phytologist, <https://nph.onlinelibrary.wiley.com/doi/full/10.1111/nph.15468>). In this earlier paper, we described syndromes for species with known pollinators and then performed extensive analyses including objective Random Forest classification algorithms to sort species with unknown pollinators into syndromes. We tested the predictive power of our approaches by re-classifying species with known pollinators into syndromes and found our syndromes to predict pollinators with high accuracy. Hence, we refrain from this point of criticism as weakening our study.

L275-277. The sample size varies widely among species, and is quite small in some species. Although such a factor has been considered in the analyses, a claim of the limited samplings would be appreciated in testing these hypotheses and illustrating their ideas.

We included such statements in the revised version of the manuscript.

With these clarifications and adjustments in the manuscript, we hope that we could convince you of the value and power of our study and alleviate all worries associated with our data.

Sincerely, on behalf of all authors,

Agnes Dellinger

REVIEWERS' COMMENTS:

Reviewer #2 (Remarks to the Author):

Comments for the authors:

The authors have addressed the concerns that I posed on the previous version of this manuscript well. However, I do still think that the caveats of the small sample sizes, etc, the reasons why larger sample sized cannot be obtained, and the extensive analyses the authors have done to prove the robustness of their data should be laid out more explicitly and earlier on in the manuscript. I also have some minor comments:

-L 121-123—does it make sense to add (Hyp 1) after 'developmental modularity' and (Hyps 2-5) after 'functional modularity' to help orient the reader?

-L 134—this is the first instance where the manuscript brings up the relatively small sample sizes of the flowers used in the analyses. I would use this opportunity to lay out the caveats, explain why they are insurmountable, and to detail the analyses that you have done to prove the robustness of your dataset. You have valid reasons why larger sample sizes are unattainable, and you have done extensive analyses to ensure that the data you present is robust. I believe it strengthens the manuscript to be up front and transparent about this.

-L 162—how did you assess different rates of morphological evolution? Are these analyses present in the Methods? I may have missed them. It would be beneficial to add slightly more information to the text here so that readers can go look for more detail in the Methods if they are interested.

-L 170-200—Flower shape evolution in Merianieae. This section kind of comes out of nowhere. Perhaps add a sentence or two to the introduction to state that you also evaluate 3D floral morphometric evolution in Merianieae. I would also start this section of the Results with what questions you address in this section.

-L 183-187—I really like that you included an explanation of the OU model. Could you also include something similar for the Lambda model and how it's different from OU in L 186?

-L 190—were regime shifts calculated for PC1 and PC2 independently or together?

-L 296—consider changing 'The first are modifications...' to 'The first entails modifications...'

-L 307-308—do you mean mammalian vertebral column? It will be clearer to add 'vertebral' if so.

-L 312-321—This is an important inclusion and addresses many of the concerns that I posed with the previous version. :)

-L 321—consider using 'as robust' instead of 'are robust'

-L 331-334—I think it may be important to include more information about the range of sample sizes for the species included. You could also acknowledge the caveats of your sample sizes and why they are insurmountable here.

-L 349-368—Which aspect of the manuscript does this connect to? Which questions are you answering with these analyses?

-L 370—Give a range of sample sizes as well as the mean

-L 438—'visualiz' should be 'visualize'

-L 501—how many species with only one specimen?

-L 521—I would add a concluding sentence that promotes the robustness of your dataset despite the caveats of the small sample sizes. Something with the same message of what you have written

in L 318-321.

-Figure 3—In the Fig 3 key, Mixed-vertebrate is still salmon, not red

-L 758, Table 1—Perhaps include that p-values in italics are significant? Possibly consider bolding them as well to make it easy for the reader to see at a glance.

Supplemental:

-Figure S1—mixed-vertebrate is still salmon not red in the key

Reviewer #3 (Remarks to the Author):

The revision has clarified all concerns raised by me and the other two reviewers. I am satisfied with their explanations for the concept difference between “pollination syndromes” and floral modularity, and for the issue of limited sample sizes in some species.

As I understand, the authors have nicely addressed all comments from the three reviewers. The new version includes data re-analyses and more concise writings, producing a compelling study with general interests in the evolution of flowers.

Reading the revision is really enjoyable. I have no more comments. The use of tomography-based 3-D flower models will open an avenue for further study on adaptation of floral traits. In Abstract, the first mentioned the technology 3-D is in brief, but writing in full “3-dimensional” would be better. In the Results section, there are more than 10 times of “we found” at the beginning of sentences. I like this style of writing too, but just feel that there are too many “we found”.

On Figure legend: *M. aurata* spelling out *Meriania* and italic.

In the following, I provide my point-by-point responses to the requests raised by the reviewers, my responses are highlighted in green.

Reviewer #2 (Remarks to the Author):

Comments for the authors:

The authors have addressed the concerns that I posed on the previous version of this manuscript well. However, I do still think that the caveats of the small sample sizes, etc, the reasons why larger sample sized cannot be obtained, and the extensive analyses the authors have done to prove the robustness of their data should be laid out more explicitly and earlier on in the manuscript. I also have some minor comments:

I have now included a paragraph in the beginning of the Results section specifying the problems associated with our limited sample size and our attempts to overcome these problems (l 132 – 141). I hope this satisfies your request.

-L 121-123—does it make sense to add (Hyp 1) after ‘developmental modularity’ and (Hyps 2-5) after ‘functional modularity’ to help orient the reader?

Yes, adjusted.

-L 134—this is the first instance where the manuscript brings up the relatively small sample sizes of the flowers used in the analyses. I would use this opportunity to lay out the caveats, explain why they are insurmountable, and to detail the analyses that you have done to prove the robustness of your dataset. You have valid reasons why larger sample sizes are unattainable, and you have done extensive analyses to ensure that the data you present is robust. I believe it strengthens the manuscript to be up front and transparent about this.

Exactly here I have added some more information on why we could not sample more and how our sample size is limited and how we approached these problems (l 132- 141).

-L 162—how did you assess different rates of morphological evolution? Are these analyses present in the Methods? I may have missed them. It would be beneficial to add slightly more information to the text here so that readers can go look for more detail in the Methods if they are interested.

You are right, the test I used (`compare.mult.evol.rates` from the `geomorph` R-package) was not specified in the methods; I have hence included explanations of the test in the Results and Methods section.

-L 170-200—Flower shape evolution in Merianieae. This section kind of comes out of nowhere. Perhaps add a sentence or two to the introduction to state that you also evaluate 3D floral morphometric evolution in Merianieae. I would also start this section of the Results with what questions you address in this section.

I have adjusted the first paragraph in this results section (l 179ff) and also indicated in the introduction, that our sampling approach enables us to assess whether species converge in 3D shape space (l 73f).

-L 183-187—I really like that you included an explanation of the OU model. Could you also include something similar for the Lambda model and how it's different from OU in L 186?

Yes, I added a sentence explaining what the Lambda model does.

-L 190—were regime shifts calculated for PC1 and PC2 independently or together?

They were calculated independently.

-L 296—consider changing 'The first are modifications...' to 'The first entails modifications...'

Adjusted.

-L 307-308—do you mean mammalian vertebral column? It will be clearer to add 'vertebral' if so.

Adjusted.

-L 312-321—This is an important inclusion and addresses many of the concerns that I posed with the previous version. :)

Thank you :-)

-L 321—consider using 'as robust' instead of 'are robust'

Adjusted.

-L 331-334—I think it may be important to include more information about the range of sample sizes for the species included. You could also acknowledge the caveats of your sample sizes and why they are insurmountable here.

I have added the sample sizes and more background information on difficulties associated with sampling (343-347).

-L 349-368—Which aspect of the manuscript does this connect to? Which questions are you answering with these analyses?

We constructed a Bayesian phylogeny for our analyses of shape evolution and modularity across Merianieae and I hence detail how we constructed the phylogeny. I included an opening sentence so that the reader knows what we used the phylogeny for. Also, we ran ancestral state reconstructions of pollination syndromes to show that bee pollination is indeed ancestral (Figure 2A) and to support painting branches to demonstrate regime shifts in association with pollinator shifts in Figure 4; I have also specified this in the text (l 382f).

-L 370—Give a range of sample sizes as well as the mean

Adjusted.

-L 438—'visualiz' should be 'visualize'

Adjusted.

-L 501—how many species with only one specimen?

15; adjusted in the text.

-L 521—I would add a concluding sentence that promotes the robustness of your dataset despite the caveats of the small sample sizes. Something with the same message of what you have written in L 318-321.

I added a concluding sentence (l 543-546).

-Figure 3—In the Fig 3 key, Mixed-vertebrate is still salmon, not red

Adjusted.

-L 758, Table 1—Perhaps include that p-values in italics are significant? Possibly consider bolding them as well to make it easy for the reader to see at a glance.

Adjusted.

Supplemental:

-Figure S1—mixed-vertebrate is still salmon not red in the key

Adjusted.

Reviewer #3 (Remarks to the Author):

The revision has clarified all concerns raised by me and the other two reviewers. I am satisfied with their explanations for the concept difference between “pollination syndromes” and floral modularity, and for the issue of limited sample sizes in some species.

As I understand, the authors have nicely addressed all comments from the three reviewers. The new version includes data re-analyses and more concise writings, producing a compelling study with general interests in the evolution of flowers.

Reading the revision is really enjoyable. I have no more comments. The use of tomography-based 3-D flower models will open an avenue for further study on adaptation of floral traits.

In Abstract, the first mentioned the technology 3-D is in brief, but writing in full “3-dimensional” would be better.

Adjusted.

In the Results section, there are more than 10 times of “we found” at the beginning of sentences. I like this style of writing too, but just feel that there are too many “we found”.

I exchanged some of the “we found” with other phrases.

On Figure legend: *M. aurata* spelling out Meriania and italic.

Adjusted.